# The Impact of Sequential Therapies after First-Line Systemic Therapies in Unresectable Hepatocellular Carcinoma

**DOI:** 10.3390/jcm13092612

**Published:** 2024-04-29

**Authors:** Shou-Wu Lee, Teng-Yu Lee, Sheng-Shun Yang, Yi-Jie Huang, Yen-Chun Peng

**Affiliations:** 1Division of Gastroenterology and Hepatology, Department of Internal Medicine, Taichung Veterans General Hospital, Taichung 40705, Taiwanericest429@vghtc.gov.tw (S.-S.Y.);; 2School of Medicine, Chung Shan Medical University, Taichung 40201, Taiwan; 3Department of Post-Baccalaureate Medicine, College of Medicine, Chung Hsing University, Taichung 40227, Taiwan; 4School of Medicine, National Yang Ming Chiao Tung University, Taipei 11221, Taiwan; 5Ph.D. Program in Translational Medicine, Chung Hsing University, Taichung 40227, Taiwan; 6Institute of Biomedical Sciences, Chung Hsing University, Taichung 40227, Taiwan

**Keywords:** atezolizumab/bevacizumab, hepatocellular carcinoma, lenvatinib, sequential systemic therapy, sorafenib

## Abstract

**Background:** The therapeutic options for hepatocellular carcinoma (HCC) have greatly expanded recently, and current first-line therapies include sorafenib, lenvatinib, and atezolizumab-bevacizumab. The aim of this study was to investigate the therapeutic efficacy of sequential systemic treatments after progressing to the first-line agent in patients with unresectable HCC. **Methods:** Data were collected from subjects with HCC, BCLC stage B or C, who received first-line sorafenib, lenvatinib, or atezolizumab-bevacizumab from September 2020 to December 2022. The patients who progressed after first-line therapy were evaluated according to individual clinical status in order to decide whether or not to accept sequential therapy. The clinical baseline characteristics and overall survival (OS) of enrolled patients were collected and further analyzed. **Results:** Among the 127 enrolled patients, percentage of sequential therapy was 67.9%, 21.6%, and 37.5% in those with tumor progression after first-line sorafenib, lenvatinib, or atezolizumab-bevacizumab, respectively. Acceptance of sequential therapy (HR 0.46, *p* = 0.041) and presentation of ALBI grade I (HR 0.36, *p* = 0.002) had a significantly positive impact on OS. Pre-treatment ALBI grade had a significant impact on the decision to accept sequential therapy in patients with progressed HCC. **Conclusions:** The patients who were able to undergo sequential therapy had a better survival outcome compared to those who received only one agent, and the pre-treatment ALBI level might be regarded as a cornerstone tool to assess survival outcomes in patients undergoing treatment for HCC.

## 1. Introduction

Hepatocellular carcinoma (HCC) is the most frequent primary liver tumor and represents an important global health challenge [1]. HCC is often associated with a known underlying etiology, such chronic hepatitis B virus (HBV) infection, chronic hepatitis C (HCV) infection, alcohol intake, and aflatoxin exposure. Recently, the incidence of HCC associated with metabolic dysfunction-associated steatotic liver disease (MASLD) has been increasing [2].

Currently in Taiwan, the United States, and Japan, less than two-thirds of patients with HCC belong to early stage and possible candidates for curative treatments, while more than one-third of patients must receive non-curative treatments [3,4], such as systemic therapy. The systemic therapeutic options for this neoplasm have greatly expanded during the past decade, which has seen the approval of various choices for both first and second-line therapies [5].

Among the first-line therapies, four treatments options are available, which include the two tyrosine kinase inhibitors (TKIs), sorafenib and lenvatinib; a combination of the anti-programmed cell death ligand 1 (PD-L1) atezolizumab with the anti-vascular endothelial growth factor (VEGF) bevacizumab; and a combination of a single primary dose of the anti-cytotoxic T lymphocyte antigen 4 (CTLA-4) tremelimumab with the anti PD-L1 durvalumab, based on successful phase 3 studies [6,7,8,9]. These two combination agents of atezolizumab-bevacizumab and tremelimumab-durvalumab displayed a better OS compared to sorafenib in patients with unresectable HCC, so they were preferred as the first-line therapeutic option.

However, until now, there are no randomized clinical trials that have directly compared these combination agents with Lenvatinib [2].Furthermore, there are conditions that limit the prescription of immunotherapy in clinical practice, including individuals with autoimmune diseases or those that require chronic systemic immunosuppression, therefore lenvatinib and sorafenib remain the alternative first-line therapy of choice. Lastly, the combination of durvalumab and tremelimumab is not yet reimbursed in many countries, and based on the better tumor response rate, the more frequently used first-line options nowadays are lenvatinib and atezolizumab-bevacizumab.

With regard to second-line treatments, regorafenib, cabozantinib, and ramucirumab are the currently approved second-line therapies for patients with preserved liver function who experience progression during sorafenib. The RESORCE and CELESTIAL trials led to the approval of regorafenib and cabozantinib, respectively, in patients who have progressed to first-line sorafenib [10,11]. Ramucirumab, an anti-VEGF receptor, is also available as a second-line option in patients with baseline α-fetoprotein (AFP) levels greater than 400 ng/mL and who have progressed to sorafenib, based on results of the REACH-2 trial [12]. The use of immunotherapeutic drugs, anti PD-1 pembrolizumab and the combination of anti-CTLA4 ipilimumab with anti PD-1 nivolumab, in the second-line setting after sorafenib, was approved based on two phase II trials, KEYNOTE-224 and CheckMate 040, respectively [13,14]. To date, all approved second-line randomized phase 3 studies were conducted in patients who progressed to the first-line option with sorafenib.

According to previous real-world studies, gradual improvements in the survival outcomes of patients with HCC receiving systemic therapy over the last decade were noted, and the use of sequential treatment after tumor progression was beneficial to prolong survival outcomes [15,16].

The aim of this study was to investigate the therapeutic efficacy of sequential treatments after progressing to the currently available first-line systemic therapy in patients with unresectable HCC, and to analyze variables associated with the ability to receive sequential treatments.

## 2. Materials and Methods

### 2.1. Patient Enrollment

Data were collected from subjects with HCC, Barcelona Clinic Liver cancer (BCLC) classification stage B or stage C, as diagnosed according to the American Association for the Study of Liver Disease (AASLD) guidelines [17], who were receiving first-line treatment with sorafenib, lenvatinib, or atezolizumab-bevacizumab at Taichung Veterans General Hospital from September 2020 to December 2022. All enrolled cases presented with cirrhosis Child-Pugh stage A.

The exclusion criteria included cases diagnosed with cirrhosis Child-Pugh stage B or C, HCC BCLC stage A or D, poor performance status, lack of compliance to drugs, or loss of follow-up within the following day.

### 2.2. Data Organization

The general data of enrolled patients, including age, gender, presence of chronic HBV or HCV infection, albumin–bilirubin (ALBI) grade, BCLC stage, numbers and size of HCC, tumor with macroscopic vascular invasion (MVI) or extrahepatic spread (EHS), and serum level of alpha-fetoprotein (AFP) of each individual were recorded.

After administration of first-line therapy, the subjects were followed up in the outpatient clinic every 2 to 4 weeks. Tumor response through imaging was assessed every 4 to 8 weeks by experienced radiologists. The assessment of the tumor response was completed according to the modified RECIST (mRECIST) criteria [18], with four response categories: complete response (CR), partial response (PR), stable disease (SD), and progressive disease (PD). The patients who had CR, PR, or SD with first-line therapy were labelled as those without progressive disease.

These therapies were discontinued if obvious tumor progression was disclosed on subsequent images. These patients with tumor progression accepted or declined sequential second-line or third-line therapy for viable HCC, which was determined by an experienced hepatologist according to individual clinical status. The therapeutic agents of sequential second-line or third-line therapy were recorded. Overall survival (OS) was defined as the time from the start of first-line therapy until death or until last follow-up.

### 2.3. Statistical Analysis

Data are expressed as standard deviation of mean for each of the measured parameters. Each stratified group is expressed as a percentage of the total patient number. Statistical comparisons were made using Pearson’s chi-square test in order to compare the effects of the positive percentage of each stratified group. An independent *t*-test was used to analyze the continuous variables. A *p*-value below 0.05 was considered statistically significant. Survival analysis was carried out using the Kaplan-Meier method for univariable and multivariable analysis, and subsequently comparisons were performed with the log-rank test.

## 3. Results

### 3.1. Patient Characteristics and Therapeutic Agents

A total of 127 subjects, comprising 38, 66, and 23 cases who received first-line agents sorafenib, lenvatinib, and atezolizumab-bevacizumab, respectively, were enrolled, and their general data are listed in Table 1. The mean age was 67.1 years, and males were predominantly present (male 81.1%). There were 55 patients (43.3%) with HBV infection and 42 patients (33.1%) with HCV infection, respectively. ALBI grade 1 and 2 accounted for 48.8% and 51.2% of all the patients. The proportion of BCLC stage B and C was 15.7% and 84.3%, respectively.

The radiological responses to first-line treatment and sequential therapy of the patients with tumor progression are shown in Table 2. Among the patients receiving first-line therapy with sorafenib, lenvatinib, atezolizumab-bevacizumab, 10 (26.3%), 29 (43.9%), and 7 (30.4%) had a non-progressive tumor status. Among 28, 37, and 16 individuals with tumor progression after first-line therapy with sorafenib, lenvatinib, atezolizumab-bevacizumab, 19 (50.0%), 8 (12.1%), and 6 (26.1%) cases had sequential second-line therapy. Most patients with first-line sorafenib received regorafenib (14 cases) as the second-line therapy, followed by ramucirumab (2 cases), cabozantinib (2 cases), and atezolizumab-bevacizumab (1 case). Those taking first-line lenvatinib received nivolumab (2 cases), sorafenib (1 case), atezolizumab-bevacizumab (1 case), regorafenib (1 case), ramucirumab (1 case), cabozantinib (1 case), and pembrolizumab (1 case) as the second-line agent. The patients with first-line atezolizumab-bevacizumab received sorafenib (2 case), lenvatinib (2 cases), and nivolumab–ipilimumab (2 cases) as the sequential second-line option. There were 4 patients who received third-line therapy, two with lenvatinib and two with pembrolizumab.

### 3.2. Variables Associated with Receiving Sequential Therapy after Tumor Porgression

A comparison of subjects who received and did not receive sequential therapy after tumor progression is shown in Table 3. Significantly more patients receiving sequential therapy were classified as pre-treatment ALBI grade 1 than those who did not receive sequential therapy (69.7% vs. 35.4%, *p* = 0.002). Other clinical variables, including age, gender, post-treatment ALBI grade, BCLC stage, appearance of MVI or EHS, the size or number of tumors, and serum level of AFP, showed no significant differences between these two groups.

### 3.3. Patients’Overall Survival and the Associated Factors

Further analysis of OS when stratified by each clinical variable is shown in Table 4. Acceptance of sequential therapy (HR 0.46, 95% CI 0.22–0.97, *p* = 0.041) and presentation of baseline ALBI grade I (HR 0.36, 95% CI 0.19–0.68, *p* = 0.002) had significantly positive impacts when adjusted in the multivariable analysis. Lower serum level of AFP (<400 mg/mL) had a significantly positive impact on OS initially, but the significance was lost after adjustment in the multivariable analysis (HR 0.65, 95% CI 0.37–1.13, *p* = 0.128). In contrast, other variables, including age, gender, viral hepatitis, BCLC stage, appearance of MVI or EHS, and different first-line options, had no significant impact on survival outcomes.

As shown in Figure 1, the median OS (95% CI) of those with non-progressive disease, those who received sequential therapy, and those without sequential therapy were 19.4 months (15.8–22.1), 17.7 months (11.3–22.7), and 11.4 months (8.2–13.8), respectively. The survival outcome was similar between the patients with non-progressive disease and those with sequential therapy (HR 0.69, 95% CI 0.21–1.18, *p* = 0.112), but individuals without sequential therapy had a significant poorer OS compared with the other two groups of patients (HR 0.40, 95% CI 0.19–0.82, *p* = 0.0013; HR 0.41, 95% CI 0.20–0.83, *p* = 0.0013).

## 4. Discussion

HCC is one of the leading causes of cancer-related mortality worldwide, and the systemic therapy strategy for patients with unresectable HCC begins with sorafenib as the first-line systemic therapy since 2007, based on a prolongation of median OS from 7.9 to 10.7 months (HR 0.69, C.I. 0.55 to 0.87, *p* < 0.001) and radiologic progression from 2.8 to 5.5 months (*p* < 0.001), approved by the results of the SHARP trial [6]. According to the REFLECT study in 2018, lenvatinib was shown to be non-inferior to sorafenib and was provided as another standard first-line systemic therapy [7]. In the phase III IMbrave150 trial, combined atezolizumab-bevatezolizumab-bevacizumab was shown to have superior activity in first-line treatment compared to sorafenib, with a statistically significant improvement in OS at 12 months (67.2% vs. 54.6%) [8]. In an updated analysis, the combination therapy of atezolizumab-bevacizumab was also confirmed to obtain a better outcome both in terms of a median OS from 13.4 to 19.2 months and median progression-free survival (PFS) from 4.3 to 6.9 months [19], and it has become the new first-line standard of care [2]. Recently, the phase III HIMALAYA trail, comparing the combination of tremelimumab plus durvalumab to sorafenib in patients naive to systemic therapy, found it to be superior in terms of median OS from 13.7 to 16.4 months (HR 0.78; 96% CI, 0.65–0.92; *p* = 0.0035) [9].The updated data from the four-year study are available, showing that the efficacy and safety of tremelimumab–durvalumab are consistent with those of the primary analysis, in particular the OS rates at 36 and 48 months are 30.7% and 25.2% for tremelimumab-durvalumab vs.19.8% and 15.1% for sorafenib [20].

To date, research efforts investigating sequential therapy in HCC after sorafenib failure have proven this from prospective clinical studies including TKIs regorafenib and cabozantinib, and VEGFR-2 inhibitor ramucirumab. Regorafenib, having a stronger action on the VEGF pathway, was the first agent to demonstrate an OS benefit over placebo after sorafenib failure, 10.6 vs. 7.8 months (HR 0.62, *p* < 0.001), respectively, within the phase III RESORCE trial [10]. Further prospective data on sorafenib–regorafenib sequential systemic treatment were provided using a post hoc analysis of the III RESORCE study, which showed a median OS of 26.0 months [21]. Cabozantinib, a TKI with multiple targets, approved effectiveness in the CELESTIAL phase III trial, with a significant improvement in the median OS over placebo from 8.0 to 10.2 months [11]. Ramucirumab is a monoclonal antibody for VEGFR2, demonstrating its superiority over the placebo in terms of median OS from 7.3 to 8.5 months for patients with serum AFP levels >400 ng/mL, proven by the REACH-2 trial [12].

The use of immunotherapeutic drugs, anti PD-1 pembrolizumab and the combination of anti-CTLA4 ipilimumab with anti PD-1 nivolumab, in the second-line setting after sorafenib, was approved based on two phase II trials, KEYNOTE-224 and CheckMate 040, respectively [13,14].Unfortunately, immune checkpoint inhibitor mono-therapies have failed to show a statistically significant improvement in OS in the first- and second-line setting in the further phase III trials [22,23].

Only one third of patients treated with lenvatinb in the phase III REFLECT trial received second-line therapy, and the post hoc analysis of patients undergoing second-line therapy after prior lenvatinib reported a median OS of 20.8 months [24]. However, as for second-line therapies after lenvatinib, to date there are no direct data available from phase III randomized clinical trials. Similarly, as in those of lenvatinib, there are no phase III studies associated with therapeutic regimens approved after atezolizumab-bevacizumab.

Our results showed a higher rate of accepting second-line therapy in the subjects receiving sorafenib as a first-line therapy (67.9%), and lower in those taking lenvatinib (21.6%) or atezolizumab-bevacizumab as a first-line therapy (37.5%). The lower percentage of sequential therapy after first-line lenvatinib or atezolizumab-bevacizumab might be due to the lack of prospective phase 3 studies confirming the effectiveness of sequential therapy agents. Another reason for rejecting sequential therapy might be related to socio-economic issues, such as the high cost of second-line agents that patients could not afford.

Real-world data in Japan study showed that 65.8–78.4% of patients would receive second-line therapy after sorafenib. Favorable indicators identified included a good Child-Pugh score, high albumin level, and a good ALBI grade, indicating that good liver function is the most important factor [25]. A retrospective study that enrolled patients from seven institutions in Japan reported the percentage of patients receiving second-line agent was 41.7% in advanced HCC with lenvatinib as the first-line agent. Among the second-line therapies, the patients taking regorafenib had a longer PFS than those on sorafenib (3.2 months vs. 1.8 months) [26]. An international multiple-center study collected data on patients who progressed to lenvatinib (n = 917) or atezolizumab–bevacizumab (n = 464) as the first-line treatment for advanced HCC, but found no statistical difference in OS between these two groups (median OS, lenvatinib 20.6 months vs. atezolizumab-bevacizumab 15.7 months, HR = 0.80, *p* = 0.12). After first-line lenvatinib, there were no statistical differences in survival outcome among second-line agents. After first-line atezolizumab–bevacizumab, lenvatinib as the second-line agent had a significant benefit of OS (HR:0.50, *p* < 0.01) [27].

One study with 877 patients with advanced HCC in Japan accepted TKIs as first-line and sequential therapy and found the avidity of multiple TKIs steadily improved the prognosis of these individuals [15]. Another retrospective study, including 336 Japanese patients who began systemic therapy, found the median OS was significantly prolonged, which was considered to be associated with the increased proportion of those who were administered sequential therapy. The variables of modified ALBI grade 2b or 3, multiple tumor numbers, extrahepatic metastasis, and serum AFP level over 400 ng/mL were the strongest factors associated with poor survival outcomes [16].

In our study, patients who were able to undergo sequential therapy showed a similar survival outcome with those without progressive tumor status (median OS, 17.7 months vs. 19.4 months, HR 0.69, *p* = 0.112), but had a significant longer survival outcome compared to those who received only one agent(median OS, 17.7 months vs. 11.4 months, HR 0.41, *p* = 0.013). Further analysis found patients who received sequential systemic therapy had significantly better ALBI level at baseline. Thus, the ALBI level might be regarded as a cornerstone tool to assess survival outcomes before the initiation of systemic first-line therapy against HCC. Liver function might have deteriorated during the initial treatment for HCC, but post-progression AIBI grade had no significant impact on the decision to use sequential treatment or not in our study.

There were several limitations in our study. First, this study used a retrospective design and was conducted as a single tertiary care center. Selection bias may therefore have existed. Second, the number of enrolled cases was small, so the therapeutic benefit of different sequential agents could not be analyzed. Third, patients who were intolerant to first-line agents were excluded from our study, and that might have led to an underestimation of the prevalence of sequential therapy. Furthermore, socio-economic issues, for example, the cost of sequential therapy agents might influence the patients’ willingness to accept them. Fourth, the therapeutic doses of oncospecific treatments were not collected, and this may have affected survival outcomes. Further prospective research that includes analysis of a larger number of variables is therefore warranted.

## 5. Conclusions

The patients who were able to undergo sequential therapy had a better survival outcome compared to those who received only one agent, and the pre-treatment ALBI level might be regarded as a cornerstone tool to assess survival outcomes in patients undergoing systemic treatment for HCC.

## Figures and Tables

**Figure 1 jcm-13-02612-f001:**
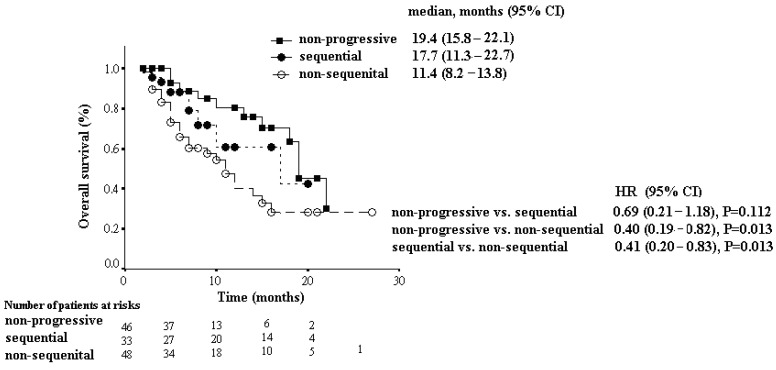
Overall survival in patients with non-progressive disease, those who received or did not receive sequential therapy.

**Table 1 jcm-13-02612-t001:** Baseline characteristics in patients who underwent first-line therapies.

		Total (N = 127)		
		M ± SD	N	%
Age (years)		67.1 ± 10.7		
	≧65		80	(63.0%)
	<65		47	(37.0%)
Gender (male)			103	(81.1%)
Hepatitis	HBV		55	(43.3%)
	HCV		42	(33.1%)
ALBI grade	1		62	(48.8%)
	2		65	(51.2%)
BCLC stage	B		20	(15.7%)
	C		107	(84.3%)
MVI			72	(56.7%)
EHS			49	(38.6%)
HCC size		5.8 ± 4.7		
HCC N ≧ 3			70	(55.1%)
AFP (×10^3^ ng/mL)		10.5 ± 40.4		
AFP(ng/mL)				
	≧400		49	(38.6%)
	<400		78	(61.4%)

Abbreviations: AFP, alpha-fetoprotein; ALBI grade, albumin-bilirubin grade; BCLC stage, Barcelona Clinic Liver cancer stage; EHS, extrahepatic spread; HBV, Hepatitis B; HCC, hepatocellular carcinoma; HCV, Hepatitis C; M, mean; MVI, macroscopic vascular invasion; N, numbers; SD, standard derivation.

**Table 2 jcm-13-02612-t002:** The radiological responses to first-line treatment and sequential therapy of patients with tumor progression.

	Total Number (N = 127)					
	Sorafenib (N = 38)		Lenvatinib (N = 66)		Atezolizumab-Bevacizumab (N = 23)	
	N	%	N	%	N	%
non-progressive disease	10	(26.3%)	29	(43.9%)	7	(30.4%)
progressive disease	28	(74.4%)	37	(56.1%)	16	(69.6%)
second-line therapy	19	(50.0%)	8	(12.1%)	6	(26.1%)
sorafenib			1		2	
lenvatinib					2	
atezolizumab-bevacizumab	1		1			
regorafenib	14		1			
ramucirumab	2		1			
cabozantinib	2		1			
nivolumab			2			
nivolumab-ipilimumab					2	
pembrolizumab			1			
third-line therapy	2	(5.2%)	1	(1.5%)	1	(4.3%)
lenvatinib	2					
pembrolizumab			1		1	

Abbreviations: N, number.

**Table 3 jcm-13-02612-t003:** The association of clinical variables and the patients with or without sequential therapy.

		Total Number with Tumor Progression (N = 81)						
		Sequential Therapy (N = 33)			No Sequential Therapy (N = 48)			
					M ± SD	N	%	*p*-Value
Age (years)		67.5 ± 11.6			67.1 ± 10.9			0.891
	≧65 y/o		21	(63.6%)		32	(66.7%)	0.778
	<65 y/o		12	(36.4%)		16	(33.3%)	
Gender (male)			28	(84.8%)		40	(83.3%)	0.855
Hepatitis	HBV		18	(54.5%)		21	(43.8%)	0.339
	HCV		9	(27.3%)		12	(25.0%)	0.810
Pre-treatment ALBI grade								
	1		23	(69.7%)		17	(35.4%)	0.002
	2		10	(30.3%)		31	(64.6%)	
Post-progression ALBI grade								
	1		10	(30.3%)		7	(14.6%)	0.088
	2		23	(69.7%)		41	(85.4%)	
BCLC stage	B		6	(18.2%)		7	(14.6%)	0.665
	C		27	(81.8%)		41	(85.4%)	
MVI			16	(48.5%)		30	(62.5%)	0.211
EHS			14	(42.4%)		18	(37.5%)	0.656
HCC size		5.9 ± 3.6			7.1 ± 5.4			0.162
HCC N ≧3			20	(60.6%)		28	(58.3%)	0.838
AFP (×10^3^ ng/mL)		10.0 ± 32.1			18.0 ± 59.3			0.481
AFP (ng/mL)	≧400		11	(33.3%)		21	(43.8%)	0.346
	<400		22	(66.7%)		27	(56.3%)	

Abbreviations: AFP, alpha-fetoprotein; ALBI grade, albumin–bilirubin grade; BCLC stage, Barcelona Clinic Liver cancer stage; EHS, extrahepatic spread; HBV, Hepatitis B; HCC, hepatocellular carcinoma; HCV, Hepatitis C; M, mean; MVI, macroscopic vascular invasion; N, number; SD, standard derivation.

**Table 4 jcm-13-02612-t004:** The strength of association between overall survival and clinical variables.

	Univariable Analysis			Multivariable Analysis		
Overall Survival	HR	(95% CI)	*p*-Value	HR	(95% CI)	*p*-Value
Age ≤ 65 (vs. >65 years old)	0.83	(0.46–1.50)	0.530			
Gender male (vs. female)	1.32	(0.62–2.81)	0.475			
HBV (vs. non-HBV)	0.96	(0.55–1.66)	0.884			
HCV (vs. non-HCV)	0.96	(0.54–1.68)	0.880			
ALBI grade 1 (vs. 2)	0.31	(0.17–0.57)	0.001	0.36	(0.19–0.68)	0.002
BCLC stage C (vs. B)	0.68	(0.32–1.45)	0.324			
MVI (vs. non-MVI)	1.77	(0.99–3.16)	0.053			
EHS (vs. non-EHS)	1.30	(0.72–2.37)	0.383			
AFP ≤ 400 (vs. >400 ng/mL)	0.54	(0.31–0.94)	0.030	0.65	(0.37–1.13)	0.128
First-line (sorafenib vs. lenvatinib)	0.91	(0.48–1.71)	0.768			
First-line (sorafenib vs. atezolizumab-bevacizumab)	0.71	(0.29–1.72)	0.442			
First-line (lenvatinib vs. atezolizumab-bevacizumab)	1.01	(0.47–2.15)	0.997			
Sequential therapy (vs. no sequential)	0.40	(0.20–0.83)	0.013	0.46	(0.22–0.97)	0.041

Abbreviations: AFP, alpha-fetoprotein; ALBI grade, albumin-bilirubin grade; BCLC stage, Barcelona Clinic Liver cancer stage; CI, confidence interval; EHS, extrahepatic spread; HBV, Hepatitis B; HCV, Hepatitis C; HR, hazard ratio; MVI, macroscopic vascular invasion.

## Data Availability

The data presented in this study are available on reasonable request to the corresponding author.

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
