# Peer review of "The Impact of Sequential Therapies after First-Line Systemic Therapies in Unresectable Hepatocellular Carcinoma"

_jcm, 2024, doi:10.3390/jcm13092612_

Round 1
Reviewer 1 Report
Comments and Suggestions for Authors
This article is written very briefly and gives the impression of a short message, rather than a full-length article. It is necessary to significantly expand the sections "Introduction", "Materials and Methods" and "Discussion of Results". In the “ Introduction” section, it is necessary to write in more detail about all the listed drugs, their mechanisms of action and practical use, referring to earlier works. The section "Materials and Methods" should be divided into subsections and more systematized, the obligatory section "Statistical processing". Since there are few methods and results obtained, it is then necessary to discuss them in more detail, referring to other works.
Author Response
Thanks for your command. We have rewritten our manuscript with expand the sections "Introduction", "Materials and Methods" and "Discussion of Results", adding more detail about all the listed drugs, dividing into subsection of the section "Materials and Methods", discussing more detail information in the section "Discussion".
Reviewer 2 Report
Comments and Suggestions for Authors
The conclusion that baseline ALBI level should be regarded as a cornerstone tool to assess survival outcomes before the initiation of systemic first-line therapy against HCC seems like a stretch. Although the P value is significant, it would be beneficial to visualize this data separately (in a different format like a bargraph) to see how the distribution of the individual values look before making this conclusion.
I am not sure how beneficial the conclusions drawn from this study are. A more effective study would be to increase the sample size and compare the efficiency of each sequential therapy against one another to determine which one preforms better.
In the introduction of the manuscript, the authors could add statements regarding the significance of the current study. Why is it important to understand the impact of sequential therapies after the failure of first-line therapies. Can this data benefit physicians/patients in any way?
One missing aspect from the discussion is an exploration of the factors influencing a patient’s decision to either adopt or reject sequential therapy. Could socio-economic circumstances play a role in this decision?
Author Response
The conclusion that baseline ALBI level should be regarded as a cornerstone tool to assess survival outcomes before the initiation of systemic first-line therapy against HCC seems like a stretch. Although the P value is significant, it would be beneficial to visualize this data separately (in a different format like a bargraph) to see how the distribution of the individual values look before making this conclusion.
Answer: Thanks for your command. In our study, the baseline ALBI grade is significantly associated with not only receiving sequential therapy after tumor progression but also survival outcomes of these patients. The emphasize had been made in the Table 3, 4 and manuscript text. Thus, to avoid duplication, we do not add another bargraph in the revised manuscript,
I am not sure how beneficial the conclusions drawn from this study are. A more effective study would be to increase the sample size and compare the efficiency of each sequential therapy against one another to determine which one preforms better.
Answer: Thanks for our command. Indeed, one of limitation of our study was a small sample size, and we have put it into the limitation paragraph.
In the introduction of the manuscript, the authors could add statements regarding the significance of the current study. Why is it important to understand the impact of sequential therapies after the failure of first-line therapies. Can this data benefit physicians/patients in any way?
Answer: Thanks for our command. We have add the more information about the impact of sequential therapies to patients with HCC into the section "Introduction".
One missing aspect from the discussion is an exploration of the factors influencing a patient’s decision to either adopt or reject sequential therapy. Could socio-economic circumstances play a role in this decision?
Answer: Thanks for our command. Indeed, socio-economic circumstances might account to patient's choices to receive sequential therapy or not. We have add this statement into the section" Discussion" and limitation paragraph.
Reviewer 3 Report
Comments and Suggestions for Authors
1) Introduction sections should be improved to highlight the significance and novelty of the study. From the current introduction, it seems the study lacks novelty and significance.
2) References used in this study are quite old. Add updated references.
3) A STROBE checklist should be provided for the study.
4) In lines 26 and 198, the ALBI level should be regarded as a cornerstone tool……. This statement has not been tested in detail. So, it must be toned down.
5) A scheme for Sequential systemic therapies by the patient over time should be shown.
6) I wonder if all patients included in the study have died.
7) Along with OS, the Course of AFP and ALBI should be shown
Comments on the Quality of English LanguageMinor editing of English language required.
Author Response
1) Introduction sections should be improved to highlight the significance and novelty of the study. From the current introduction, it seems the study lacks novelty and significance.
Answer: Thanks for our command. We have add the more information about the impact of sequential therapies to patients with HCC into the section "Introduction"
2) References used in this study are quite old. Add updated references.
Answer: Thanks for our command. We have added and update some new references in the revised manuscript.
3) A STROBE checklist should be provided for the study.
Answer: Thanks for our command. STROBE checklist is checked.
4) In lines 26 and 198, the ALBI level should be regarded as a cornerstone tool……. This statement has not been tested in detail. So, it must be toned down.
Answer: Thanks for our command. We have revised "should" to "might" in the revised manuscript.
5) A scheme for Sequential systemic therapies by the patient over time should be shown.
Answer: Thanks for our command. The scheme of sequential systemic therapies to HCC has added in the section “Introduction" and “Discussion" in the revised manuscript.
6) I wonder if all patients included in the study have died.
Answer: Thanks for our command. As shown in Figure 1, at the end-time of following, there were about 20% of patient still alive.
7) Along with OS, the Course of AFP and ALBI should be shown
Answer: Thanks for our command. As shown in Table 4, AFP and ALBI were listed and analyzed to comparing with OS of patients. Besides, "pre-treatment ALBI" and "after-treatment ALBI" were listed and analyzed to comparing the possibility to receive sequential systemic therapy or not.
Round 2
Reviewer 1 Report
Comments and Suggestions for Authors
In this form, I recommend the article for publication.
Author Response
Thank you very much.
Reviewer 2 Report
Comments and Suggestions for Authors
I appreciate that the authors have considered all comments and addressed them appropriately.
Author Response
Thank you very much.
Reviewer 3 Report
Comments and Suggestions for Authors
Reference 16 needs to be cited completely.
Author Response
Thankyou very much. The only one suggestion of reviewer had is corrected in the revised manuscript.